# Asymmetric tapered multistage solar still with optimized mass transfer equilibrium for ultrahigh water production

Wen He[1,6], Jiacheng Wang[2,6], Baiyi Chen[3,6], Yiyao Li[4], Zhongyi Fang[1], Xuan Zhou[1], Lei Zhou[4], Meng Li [2] & Xu Hou [1,4,5] ✉

Solar membrane distillation offers a highly promising and sustainable solution to the global freshwater crisis. However, its widespread practical application is currently hampered by a key challenge: Pursuing high water production. This bottleneck stems from a mismatch between the evaporation and condensation capacities in existing systems, where vapor may not be condensed in time due to insufficient condensation capacity, or the available condensation capacity may be underutilized when evaporation is inadequate. Here we show an asymmetric tapered multistage solar still that enables ultrahigh water production by introducing a design principle based on optimizing the mass transfer equilibrium between evaporation and condensation. By systematically optimizing the ratio of condensation-to-evaporation areas through a tunable mass transfer gap, the system achieves a state of ultrahigh-production equilibrium, in which evaporation and condensation processes are maximally coupled. Based on this principle, an optimized eight-stage passive solar still device is built to get a total water production of 4.32 L·m$^{-2}$·h$^{-1}$ and total $\eta_c$ of 81% under 1 kW·m$^{-2}$ illumination (with 3.1 wt% natural seawater), which ranks among the highest values reported in existing literature. It exhibits stable performance under varying light conditions and salt resistance, producing 34.2 L·d$^{-1}$ in outdoor tests.

Water scarcity is a pressing global challenge that affects billions of people, as freshwater resources continue to deplete due to climate change, population growth, and industrialization[1,2]. The development of desalination technologies is essential to address this crisis[3–6]. Although commercialized technologies like reverse osmosis and membrane distillation have seen significant development, their relatively complex processes, high costs, and environmental pollution have raised increasing concerns[7]. Recently, solar membrane

distillation has emerged as a sustainable and promising approach, utilizing abundant and renewable solar energy to produce freshwater without causing pollution[1,8]. Over the past decade, significant progresses have been made in solar membrane distillation systems, particularly with innovations in photothermal materials such as hydrogels[9–11], 3D biological structures[12–14], and magnetically controlled dynamic evaporation systems. These advances have enabled vapor production as high as 5.18 L·m$^{-2}$·h$^{-1}$ under 1-sun illumination without

[1]State Key Laboratory of Physical Chemistry of Solid Surfaces, College of Chemistry and Chemical Engineering, Xiamen University, Xiamen, China. [2]Key laboratory of Low-grade Energy Utilization Technologies & Systems, Ministry of Education, CQU-NUS Renewable Energy Materials & Devices Joint Laboratory, School of Energy & Power Engineering, Chongqing University, Chongqing, China. [3]Marine Engineering College, Fujian Provincial Key Laboratory of Advanced Marine Functional Materials, Xiamen Key Laboratory of Marine Corrosion and Smart Protective Materials, Jimei University, Xiamen, China. [4]College of Physical Science and Technology, Xiamen University, Xiamen, China. [5]Engineering Research Center of Electrochemical Technologies of Ministry of Education, Xiamen University, Xiamen, China. [6]These authors contributed equally: Wen He, Jiacheng Wang, Baiyi Chen. ✉e-mail: houx@xmu.edu.cn

additional energy input[15,16]. Despite the improvements in vapor production, the efficient conversion of vapor into water remains a bottleneck, limiting the total water production[5,10,17–19].

In recent years, multistage solar membrane distillation systems have attracted considerable attention due to their potential to recycle latent heat and enhance water production[19–22]. Existing studies have improved system performance from multiple perspectives: (i) regulating heat transfer and implementing coupled heat-mass transfer models, enabling systematic control over heat transfer, vapor diffusion, and latent heat recovery within solar stills, thus breaking the efficiency limits of conventional passive systems[19,23]; (ii) integrating multistage solar still with photovoltaic technologies to achieve water-electricity cogeneration and improve solar energy utilization[24–26]; (iii) enhancing anti-scaling capabilities, enabling stable operation at salinities up to 20 wt% while maintaining high water production[4,27–29]; and (iv) revealing how the photothermal characteristics of cover materials influence the performance in multistage solar stills, with a recent study proposing a material-thickness co-optimization paradigm that pushes efficiencies beyond 417% at low cost and high stability[30]. Additionally, studies have proposed harnessing low-grade waste heat to drive evaporation in the final stages, further improving overall system efficiency through enhanced energy reutilization[31]. These strategies have significantly enhanced the performance, stability, and integration potential of multistage solar membrane distillation systems. They have laid a solid foundation for understanding thermodynamic interactions of complex multistage configurations and provided valuable design insights for practical implementation. While these designs have greatly contributed to improving device performance, they primarily focus on optimizing either the evaporation or condensation process in isolation. To date, a systematic investigation of the mass transfer equilibrium between evaporation and condensation, which serves as the fundamental constraint governing vapor-to-water conversion, remains lacking. The mismatched evaporation and condensation capacity lead lots of vapor to fail to condense efficiently or the underutilization of condensation sites, constraining the further increase of water production and causing waste of solar energy. This limitation arises from the challenge of regulating the evaporation-condensation equilibrium to achieve the optimal mass transfer process. Therefore, it is essential to develop a principle that optimizes the evaporation-condensation equilibrium to maximize water production.

Given that evaporation and condensation are two interdependent physical processes, it is essential to establish a thermodynamically consistent framework to describe their interaction and resulting limitations in a closed solar distillation system[32]. For an open system with a photothermal evaporation surface, the input solar energy ($q_{in}$) is almost fully used for water evaporation, allowing the surface to exhibit its theoretical maximum evaporation capacity ($Q_{e, max}$). In contrast, within a closed chamber containing both evaporation and condensation surfaces, the evaporation process occurs concurrently with condensation. Here, the generated vapor cannot be removed from the system, leading to its accumulation in the chamber. As the vapor concentration and partial pressure ($P_v$) increase, the evaporation driving force (defined by the difference between $P_v$ and the saturation vapor pressure at the liquid interface) gradually decreases. When the $P_v$ approaches equilibrium with the condensation surface, the actual evaporation capacity ($Q_e$) slows until a dynamic steady state is reached, where the $Q_e$ almost equals the maximum condensation capacity ($Q_{c, max}$):

$$Q_e = Q_{c, max} \tag{1}$$

This represents an evaporation-condensation equilibrium, but not necessarily the maximum possible evaporation capacity. In fact, under this condition, $Q_e < Q_{e, max}$, indicating underutilization of solar energy due to condensation bottlenecks. Therefore, to ensure that the system

is no longer limited by condensation, it is necessary to enhance $Q_{c, max}$ in order to achieve the condition:

$$Q_{e, max} \approx Q_{c, max} \tag{2}$$

According to classical heat transfer theory, the condensation capacity can be expressed by the following relation[33],

$$Q_c = \frac{h_c A_c (T_v - T_c)}{h_{fg}} \tag{3}$$

in which $h_c$ is the heat transfer coefficient, $A_c$ represents the condensation surface area, $T_v$ is vapor temperature, $T_c$ denotes the temperature of condensation surface, and $h_{fg}$ is the vaporization enthalpy. According to Eq. (3), there are two primary ways to increase $Q_{c, max}$: (1) by maximizing the temperature difference ($\Delta T$) between $T_v$ and $T_c$, and (2) by expanding $A_c$. In a compact multistage solar membrane distillation system, it is difficult to achieve a large $\Delta T$ due to the limited mass transfer gap ($D$). As an excessively large $D$ may hinder vapor transport and thermal conduction, consequently reducing the total number of stages and impairing the system's overall water production[34]. Therefore, increasing $A_c$ is a more feasible and impactful strategy for improving $Q_{c, max}$.

However, in the existing symmetric multistage solar membrane distillation (SMSMD) systems, $A_c$ remains constant and is equal to the evaporation area ($A_e$), restricting the system's ability to regulate $Q_{c, max}$ due to the limited condensation sites (Fig. 1a). When $Q_{e, max}$ significantly exceeds $Q_{c, max}$, the mass transfer chamber rapidly reaches saturation, restricting the evaporation process and leading to a low-production equilibrium (Fig. 1b). In this condition, evaporation capacity is underutilized, resulting in suboptimal use of $q_{in}$.

Here we proposed a physical design principle: achieving ultrahigh water production by optimizing of the mass transfer equilibrium between evaporation and condensation in each stage distillation chamber of the multistage solar membrane distillation system. Specifically, we developed an asymmetric tapered multistage solar membrane distillation (ATMSMD) system, featuring a progressively enlarged structure, as shown in the right of Fig. 1a. In this system, $A_c$ is consistently larger than $A_e$ at each stage, with a ratio of $A_c \approx 1.1 \sim 1.7 A_e$. Unlike SMSMD systems, $A_c$ in the ATMSMD system can be enhanced by increasing $D$, effectively creating more condensation sites (as shown in Fig. 1b). This increased area leads to a rise in $Q_{c, max}$, enabling a more efficient vapor utilization and promoting ultrahigh-production equilibrium, where $Q_{c, max} \approx Q_{e, max}$. As continuously increasing $D$ results in a longer time required to reach equilibrium, which leads to a reduction in vapor-to-water conversion efficiency ($\eta_c$), thereby decreasing the overall water production (Fig. 1c). Therefore, it is essential to optimize the ratio between the condensation and evaporation areas in order to achieve ultrahigh-production equilibrium.

## Results
### Structure and optimization principle of single stage
The single-stage configuration of the asymmetric passive solar still (Fig. 2a) comprises four essential components: a convective blocking layer (polydimethylsiloxane film), a solar absorber (carbon nanotubes-coated aluminum foil), an evaporation layer (hydrophilic fiber), and a condensation wall (hydrophobically modified aluminum foil). The polydimethylsiloxane film exhibits a transmittance exceeding 95%, while the carbon nanotubes-coated aluminum foil achieves an absorptivity greater than 95%, ensuring sufficient heat for effective evaporation (Supplementary Fig. 1). The hydrophilic fibers are intricately interwoven (Supplementary Fig. 2a), forming micron-scaled channels that facilitate continuous water supply. Meanwhile, as shown in Supplementary Fig. 2b, c, hydrophobically modified aluminum foil

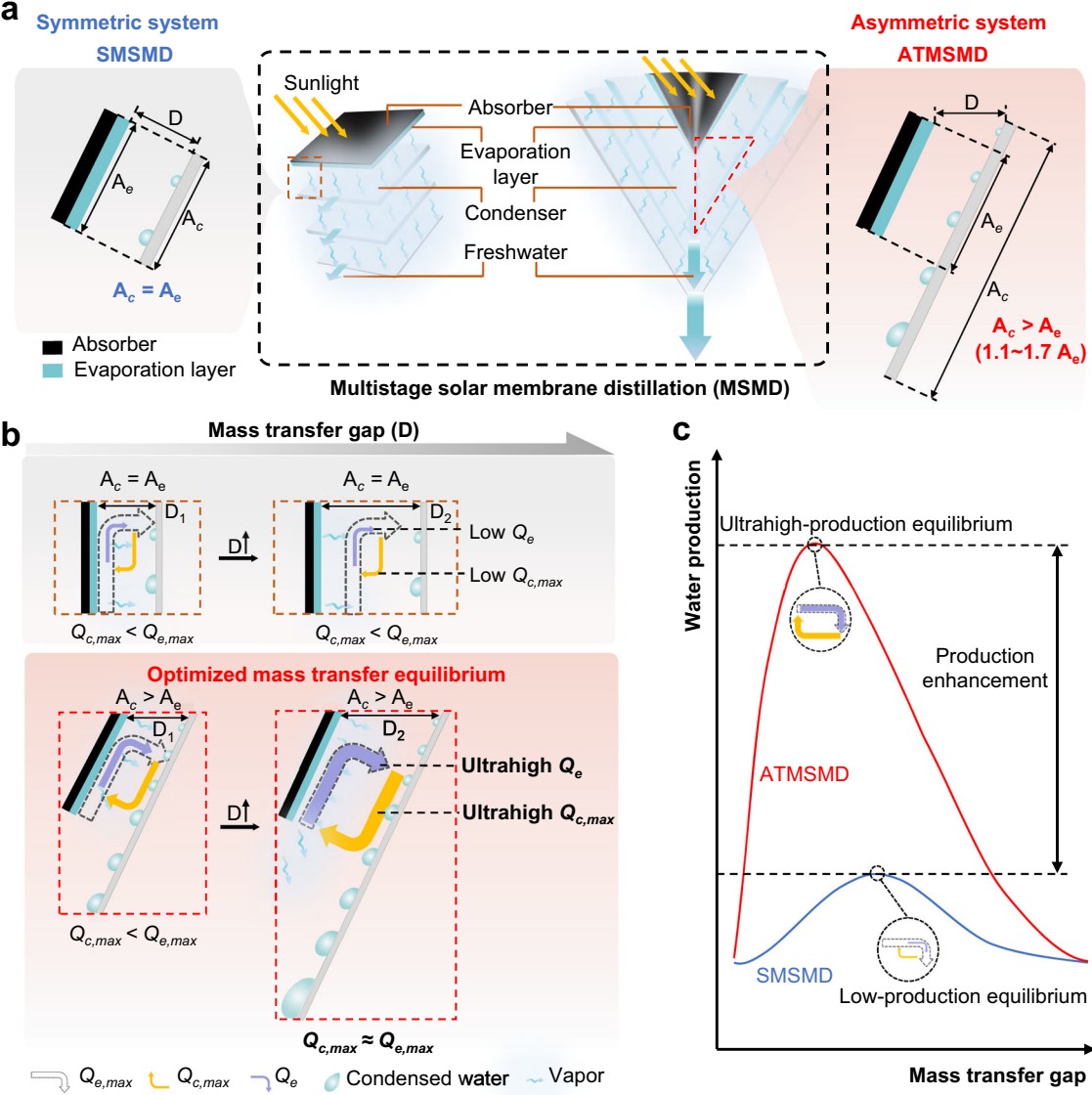

**Fig. 1 | The structure and equilibrium states of asymmetric tapered multistage solar membrane distillation system (ATMSMD). a** The structure of symmetric multistage solar membrane distillation (SMSMD) and asymmetric multistage solar membrane distillation systems. The SMSMD system exhibits a condensation area equal to its evaporation area ($A_c = A_e$), whereas the ATMSMD system exhibits an asymmetric structure, in which the condensation area is larger than the evaporation area ($A_c > A_e$, approximately 1.1–1.7$A_e$). With the same number of stages and projection area, the system achieves a higher water production than that of SMSMD system. **b** Comparison of the equilibrium states between the two systems with the same mass transfer gap. By increasing the mass transfer gap, the maximum condensation capacity ($Q_{c, max}$) of the ATMSMD system is enhanced, achieving an increase of the actual evaporation capacity ($Q_e$) and reaching ultrahigh-production equilibrium, where the $Q_{c, max}$ is approximately equal to the maximum evaporation capacity ($Q_{e, max}$). The SMSMD system shows limited improvement in $Q_{c, max}$ and low-production equilibrium. **c** Water production of ATMSMD and SMSMD systems under varying mass transfer gap (This is a concept trend diagram, which constructed based on the experimental data and the reported study[4]).

features micro-nano structures that promote dropwise condensation, enhancing the condensation site (Supplementary Fig. 3a, b). The single-stage device with a hydrophobically modified aluminum foil condensation surface achieved a 20.4% higher water production compared to the device with an unmodified aluminum foil condensation surface (Supplementary Fig. 3c).

To further optimize the performance of the single-stage asymmetric solar still, we examined the effect of the conical apex angle ($\theta$), a critical structural parameter that influences both the projected area for light absorption and the heat generation area (Supplementary Fig. 4). Accordingly, we fabricated single-stage device with $\theta$ of 60°, 90°, and 120°, and explored their evaporation and condensation capacities. As shown in Fig. 2b, the device with a 60° conical apex angle achieved the

highest vapor and water production. Furthermore, we analyzed the influence of relevant geometric parameters on the device performance, including light absorption area ($A_a$), $A_e$, $A_c$ and their corresponding ratios. Although both $A_e$ and $A_a$ increase with the $\theta$, the vapor and water production exhibit an opposite trend. We attribute this trend to the decrease in the effective evaporation area per unit illuminated surface, as indicated by the reduced $A_e/A_a$ ratio (Supplementary Fig. 5). Meanwhile, the expansion ratio of the condensation area ($A_c/A_e$) also decreases. The $A_c/A_e$ indicating limited condensation capacity and a mismatch between evaporation and condensation process ($Q_{e, max} > Q_{c, max}$), thereby limiting overall evaporation and reducing water production. In addition, a smaller $\theta$ would significantly increase the device height (e.g., reducing it from 60° to 45° increases

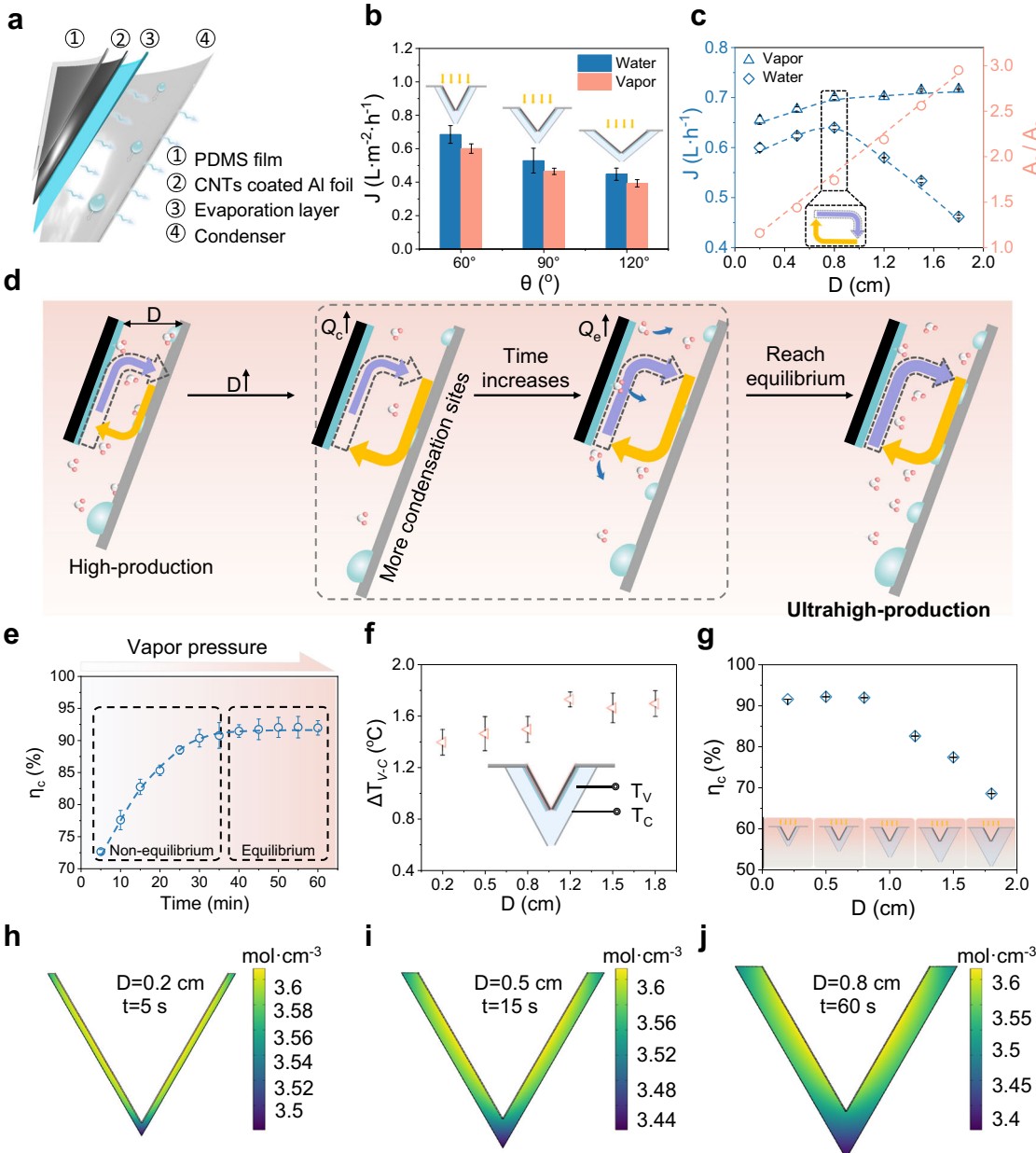

**Fig. 2 | The configuration and parameters optimization of single-stage structure. a** The configuration and materials of single-stage structure. **b** The production of single-stage structures with different conical apex angle. **c** The vapor and water production of single-stage structures with different mass transfer gap. The inset represents the point at which the system reaches an ultra-high production equilibrium. **d** The mechanism behind the increase in water production caused by the increase in mass transfer gap. $Q_e$ is the evaporation capacity, and $Q_c$ represents the condensation capacity. **e** The vapor-to-water conversion efficiency of single-stage structure with a mass transfer gap of 8 mm at different time intervals. **f** The temperature difference between vapor and condensation surface in single-stage device with different mass transfer gap. **g** The vapor-to-water efficiency of single-stage device with different mass transfer gap. **h–j** The vapor concentration distribution in the same chamber under different mass transfer gap conditions was simulated by COMSOL software. Error bars in (**b, c, e–g**) represent the s.d. ($n = 3$) and data are presented as mean values ± s.d.

the height by over 40%), which leads to greater material consumption, structural stress, and fabrication cost, while also reducing light absorption under oblique illumination. Therefore, considering the actual situation comprehensively, we adopted 60° in subsequent experiments.

We next investigated the effect of $D$ on evaporation and condensation behavior. Previous studies on SMSMD systems focused primarily on the influence of $D$ on heat transfer and energy loss, given the constant condensation area. According to the reported researches[4,23,35], $D$ values with a range of 2–20 mm is selected for optimization. In our work, we find that an optimal thickness of $D$=8 mm

exists (with $A_c/A_e = 1.7$), at which point the system reaches an ultrahigh-production equilibrium ($Q_{e,\,max} \approx Q_{c,\,max}$), as determined by experimental results (Fig. 2c). When $D$<8 mm, the mass transfer chamber reaches equilibrium rapidly, but the restricted vapor diffusion limits the evaporation capacity ($Q_{e,amx} > Q_{c,\,max}$), resulting in a low-production equilibrium. However, for $D$>8 mm, the extended vapor diffusion path delays the time required to reach mass transfer equilibrium ($Q_{e,amx} < Q_{c,\,max}$). Although vapor production continues to rise slowly with larger $D$, the extended equilibrium time ultimately leads to a reduction in water production. The mechanism underlying this phenomenon is shown in Fig. 2d, the number of condensation sites

increases with larger $D$, resulting in a rise in $Q_{c, max}$. Consequently, more vapor is required to evaporate to saturate the mass transfer chamber, leading to an increase in $Q_e$ that eventually approximated $Q_{e, max}$. However, as $D$ further increases, the additional condensation sites and $\Delta T$ delay the equilibrium, reducing the water production (Fig. 2f). Benefiting from the large condensation area of ATMSMD system and the optimized parameters, the $\eta_c$ of the single-stage asymmetric solar still reaches up to 92% (Fig. 2g).

To validate the effectiveness of the optimized mass transfer equilibrium, we tested the vapor production in open system ($J_O$) and closed chamber ($J_C$) to represent the $Q_{e, amx}$ and $Q_e$ respectively. The ratio of $J_C/J_O$ serves as a direct metric for assessing how fully the system utilizes its evaporation potential under different $D$. As shown in Supplementary Fig. 6, $J_C/J_O$ increases progressively with $D$, reaching 0.98 at the optimal $D$ of 8 mm and stabilizing around 0.99, indicating that the evaporation surface is almost close to $Q_{e, amx}$. These results demonstrate that under optimized conditions, the system operates near its physical limit of evaporation capacity, effectively achieving an ultrahigh-production equilibrium in which evaporation and condensation are maximally coupled. This high utilization rate confirms that the enlarged condensation area and tuned diffusion path eliminate the common bottleneck of incomplete vapor condensation found in symmetric designs.

To further understand the influence of $D$ on systems performance, we measured the temperature difference between the evaporation and condensation surfaces, as well as the total heat loss ($q_{loss}$), which including the conduction ($q_{cond}$), convection ($q_{conv}$), radiation ($q_{rad}$) and sidewall ($q_{side}$) heat loss at different $D$ values. The related parameters are shown in Supplementary Table 1. As shown in Supplementary Fig. 7, $\Delta T$ and $q_{loss}$ significantly increases with larger $D$. Consequently, a larger $D$ leads to a greater temperature drop across the gap, which in turn lowers the evaporation temperature and energy available for the next stage in multistage configurations. For instance, with a first-stage evaporation temperature is 46 °C, if the $\Delta T$ exceeds 5 °C, the second-stage evaporation surface may fall below 41 °C. This cumulative cooling effect will constrain the total number of effective stages and reduce the overall water production. Therefore, although increasing the condensation area by enlarging $D$ can enhance condensation capacity, $D$ cannot be increased indefinitely, as excessive $D$ leads to substantial temperature degradation and reduced evaporation performance in later stages. By combining this thermal analysis with the experimental results of water production at different $D$ values (Fig. 2c), we identify a trade-off between increasing condensation area and maintaining adequate temperature for evaporation. This integrated understanding allows us to determine the optimal $D$ based on the maximum water production. This trade-off highlights the importance of optimizing $D$ not only to achieve $Q_{e, amx} \approx Q_{c, max}$, but also to ensure thermal continuity across all stages in multistage solar still systems. Therefore, the optimal $D$ of 8 mm identified in our design reflects this thermodynamic trade-off.

Additionally, we employed COMSOL software to simulate the vapor concentration distribution in the chamber. The results of the vapor concentration distribution in the same chamber at different time indicate that the chamber gradually reaches equilibrium over time (Supplementary Figs. 8 and 9), consistent with the results shown in Fig. 2e. Meanwhile, the simulation results of vapor concentration distribution in chambers with different $D$ indicate that as $D$ increases, the time required for the chamber to reach equilibrium also increases. This implies that in chambers with smaller $D$ values, evaporation process is quickly suppressed, leading to a rapid attainment of equilibrium, ultimately resulting in low-production equilibrium (Fig. 2h–j).

## Configuration and performs of multistage device

We applied the optimization principles outlined above to an ATMSMD system. Based on previous studies[4,19,27], an initial ten-stage asymmetric

tapered solar still was fabricated, with a uniformed $D$ of 8 mm in each stage. Liquid seals are used to isolate each chamber, allowing condensed water to accumulate at the bottom of each stage and flow out through a designated collection channel. A hydrophilic region approximately 4–5 mm in height is formed at the bottom of the tapered structure by polishing the originally hydrophobically modified aluminum surface, allowing water to form a stable concave meniscus. The Laplace pressure ($\Delta p$) generated by the concave meniscus, together with the supporting force ($\mathbf{F_s}$) from the inclined surface, is balances against the gravitational force ($\mathbf{G}$), stabilizing the droplets at the pore opening of the tapered bottom (Supplementary Fig. 10). The evaporation and condensation performance of multistage device were measured using the setup illustrated in Fig. 3a. Supplementary Fig. 11 shows the vapor and water production of the ten-stage device. The water production of the ninth and tenth stages are 0.09 L m$^{-2}$ h$^{-1}$ and 0.04 L m$^{-2}$ h$^{-1}$, respectively, contributing only about 1/40 of the total water production. Based on these results, we selected the eight-stage configuration for further optimization.

The performance of the eight-stage configuration is shown in Supplementary Fig. 12. Vapor production initially increases with the number of stages, driven by the expanding $A_e$, but eventually decreased due to the diminishing evaporation temperature ($T_e$). In contrast, the water production does not mirror the trend of vapor production and progressively decreases as the number of stages increases. This decline can be attributed to the combined effects of decreasing $Q_{e, max}$ and increasing chamber volume. Larger chambers require more time to reach equilibrium, reducing the $Q_{c, max}$, which ultimately lowers the water production. By the eighth stage, the $\eta_c$ drops to 58.6%, and the water production decreases to 0.27 L·m$^{-2}$·h$^{-1}$. The total vapor and water production of the eight-stage device is 5.02 L·m$^{-2}$·h$^{-1}$ and 3.94 L·m$^{-2}$·h$^{-1}$, respectively.

Next, we analyzed the heat transfer behavior of the eight-stage device. As shown in Fig. 3b, the $q_{in}$ is absorbed by photothermal layer and convert into heat, which is subsequently conducted to evaporation surface. During this process, part of the heat dissipates through $q_{cond}$, $q_{conv}$ and $q_{rad}$. The remaining heat ($q_{e,1}$) is utilized for water vaporization in the first stage. The generated vapor condenses on the condensation surface, releasing latent heat that is transferred to the next stage for further vaporization. In the i-th stage ($i \geq 2$), the input heat $q_{in,i}$ is thus derived from the latent heat released in the previous stage's condensation. A portion of this heat is lost through $q_{side,i}$, while the remaining portion can be used for evaporation, defined as the usable energy:

$$q_{usable,i} = q_{in,i} - q_{side,i} \tag{4}$$

This formulation enables us to quantify the effective heat available for evaporation at each stage and facilitates identification of the thermodynamic constraints associated with increasing stage numbers.

As shown in Supplementary Fig. 13, $q_{usable,i}$ progressively declined with stage number due to the decreasing latent heat. For example, the second stage receives 167 W·m$^{-2}$ of usable energy, whereas the eighth stage receives only 43 W·m$^{-2}$. Consequently, beyond stage 6, $q_{usable,i}$ drops sharply, resulting in lower vapor and water production. In addition, the evaporation temperature and temperature difference between evaporation and condensation surface decrease with increasing stage number, indicating a progressive decline in evaporation and condensation capacity (Supplementary Table 2). These findings underscore the critical role of thermal confinement and stage-wise heat management in passive multistage systems and support the eight-stage configuration as an optimal balance between cascading energy utilization and thermal dissipation. The detailed calculation process is provided in the Supplementary Information, the related parameters shown in Supplementary Table 2.

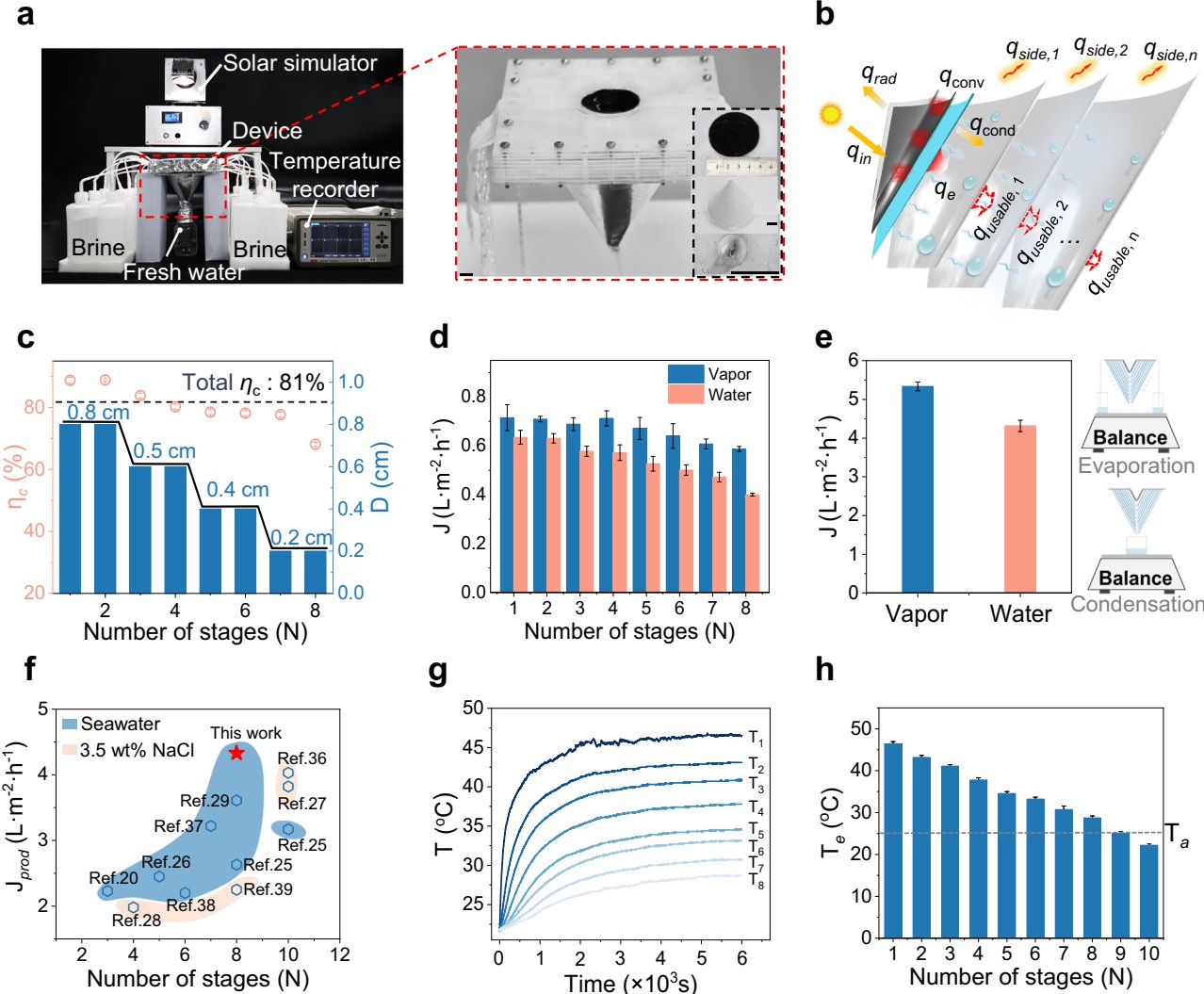

**Fig. 3 | The configuration and performance of asymmetric tapered eight-stage solar still. a** The photograph of the device with a 5 cm opening diameter. Scale bar: 1 cm. This device was used in laboratory experiments to optimize the ratio of condensation and evaporation area for each stage. The design of the larger-scale device (10 cm diameter) was based on these optimized parameters to ensure scalability and practical applicability. **b** The heat transfer process of eight-stage device. **c** The vapor-to-water conversion efficiency and mass transfer gap of each stage in eight-stage device. **d** The vapor and water production of each stage in eight-stages device. **e** The total vapor and water production of eight-stage device. Top right inset: schematic of the setup for vapor production measurements. Bottom right inset: schematic of the setup for water production measurements. **f** Water production of optimized eight-stage device compared with previously reported multistage solar membrane distillation systems. Water production is benchmarked under the treatment of natural seawater or 3.5 wt% NaCl solution. **g** The temperature distribution of each stage in optimized eight-stage device. **h** The evaporation temperature of each stage in ten-stage solar still under 1-sun illumination for 1 h. Error bars in (**c**, **d**, **e**, **h**) represent the s.d. ($n = 3$) and data are presented as mean values ± s.d.

According to the above heat distribution analysis, we then redesigned an eight-stage configuration with a progressively decreasing $D$. Considering the evaporation capacity decreases in later stages, we propose a design principle that requires the $D$ in later stages should not be larger than that in the earlier stage, to ensure that the chambers can reach equilibrium within 1 h. Therefore, we tested water production in the second stage of the two-stage devices with $D$ of 0.2, 0.4, 0.5, 0.6, and 0.8 cm. Following this optimization strategy for all eight stages, we determined the final optimal mass transfer gaps as follows: $D_1$, $D_2 = 0.8$ cm; $D_3$, $D_4 = 0.5$ cm; $D_5$, $D_6 = 0.4$ cm; and $D_7$, $D_8 = 0.2$ cm (Supplementary Fig. 14). After optimization, the chamber volumes exhibit an initial slight increase followed by a decline, closely matching the vapor production trend across the stages (Supplementary Fig. 15). The $\eta_c$ of each stage exceeded 70%, with the first five stages achieving $\eta_c$ above 80% (Fig. 3c), and the total $\eta_c$ of the device is 81%. The optimized eight-stage device demonstrated exceptional performance, culminating in a total vapor production of 5.31 L·m⁻²·h⁻¹ and a total water production

of 4.32 L·m⁻²·h⁻¹ for treating 3.1 wt% natural seawater (Fig. 3d, e). To compare the optimized eight-stage device with existing multistage solar membrane systems, we plotted diagram of the water production for existing multistage devices and our optimized eight-stage device with treating natural seawater or 3.5 wt% NaCl solution (Fig. 3f). Detailed parameters from these studies, including light intensity, feedwater type, number of stages, and water production, have been compiled in Supplementary Table 3 for direct comparison. The results show that, for devices with different numbers of stages[20,25–29,36–39], our device consistently higher water production under comparable conditions. Notably, even when compared to eight- and ten-stage configurations, our optimized device also achieves the highest water production. We attribute this improvement not merely to structural adjustments but to a mechanism-driven design principle, which emphasizes achieving optimal mass transfer equilibrium ($Q_{e, amx} \approx Q_{c, \max}$) between evaporation and condensation processes. This result supports the validity and generalizability of our proposed optimization strategy.

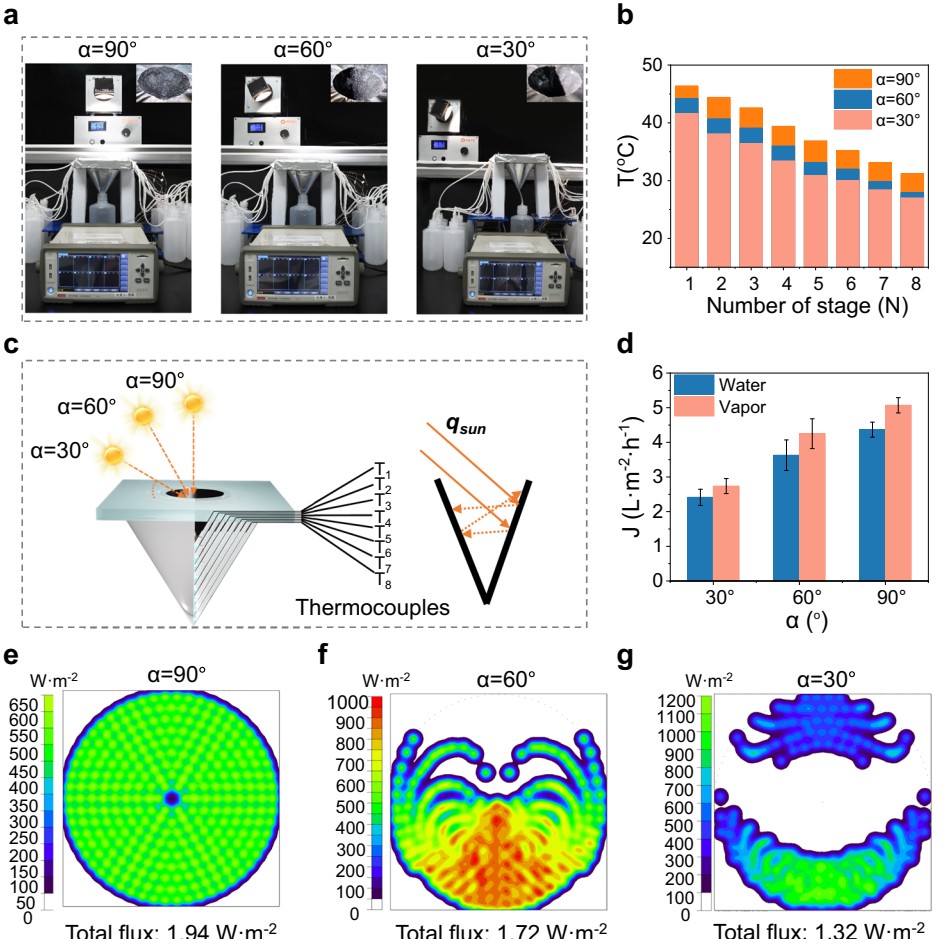

**Fig. 4 | The performance of asymmetric tapered eight-stage solar still under sunlight with different solar elevation angle. a** The photograph of testing setup under simulated sunlight with different solar elevation angle. **b** The temperature distribution of eight-stage device. **c** The diagram of test and optical path. **d** The vapor and water production under simulated sunlight with different solar elevation angle. **e–g** The optical simulation under simulated sunlight with solar elevation angle of 90°, 60°, and 30°. Error bars in (**d**) represent the s.d. ($n = 3$) and data are presented as mean values ± s.d.

The optimized device reached a stable temperature distribution within 40 min (Fig. 3g), with the first stage reached a maximum $T_e$ of 46.5 °C, while the eighth stage stabilized at 28.7 °C, meeting the minimum temperature requirement for evaporation (above ambient temperature, $T_a$). Furthermore, to investigate the feasibility of additional stages after optimizing $D$, we extended the multistage structure again to ten stages and measured $T_e$. As shown in Fig. 3h, the $T_e$ of the ninth stage is nearly equivalent to $T_a$ (25 °C), providing minimal temperature contribution to the evaporation process at this stage. Consequently, the final configuration is optimized to an eight-stage device.

**Performance of evaporation and condensation in practice**

Unlike controlled laboratory conditions, the solar elevation angle ($\alpha$) varies throughout the day in outdoor environment. To evaluate the device's performance under real-world conditions, the eight-stage device was tested under the sunlight with varying α of 30°, 60°, 90° (Fig. 4a, c and Supplementary Fig. 16). In contrast to single-pass absorption in flat structures, multiple absorption is observed in the asymmetric structure attributed to the cone design. Using the setup in Fig. 4a, we measured the water production and temperature distribution of the eight-stage device under 1-sun illumination at different α. The absorption layer of the device reaches its maximum temperature when $\alpha = 90°$ (direct overhead sunlight). As α decreases, a gradually reduction in temperature is observed. However, even at the $\alpha = 30°$, the temperature of eighth stage remains as high as 27 °C,

indicating stable photothermal performance under varying irradiation conditions (Fig. 4b).

To further investigate light absorption performance, we employed ray tracing simulations to track light paths at different α (Fig. 4e–g). The results reveal that internal reflections within the asymmetric structure enhances light absorption, even at lower solar elevation angles. Even under light at the $\alpha = 30°$, the eight-stage device achieves a water production of 2.41 L·m$^{-2}$·h$^{-1}$, demonstrating its stable performance across varying solar angles (Fig. 4d).

Furthermore, outdoor experiments were conducted to evaluate the practical applicability of the eight-stage device, focusing on water production, salt resistance, and durability. Full-day outdoor experiments demonstrate the great performance of the eight-stage device (Fig. 5a). The samples used in these experiments are 3.1 wt% seawater obtained from Baicheng Beach (Xiamen, Fujian, China). After desalination, the ion concentration in the collected water is significantly reduced compared to the original seawater, achieving a salt rejection rate of 99.9% ($B^{3+}$ of 99.1%) and meeting the standards for drinking water (Fig. 5d). Over five randomly selected days, the device shows an impressive water production of 34.2 L·d$^{-1}$ under sufficient sunlight. Furthermore, we tested the performance of the device during spring season by conducting outdoor experiments under 3 different weather conditions: sunny day, partly cloudy day, and overcast day. The results show that even in springtime, the eight-stage device achieves a daily water production of 23.3 L·m$^{-2}$·d$^{-1}$ on a sunny day, 17.8 L·m$^{-2}$·d$^{-1}$ on a

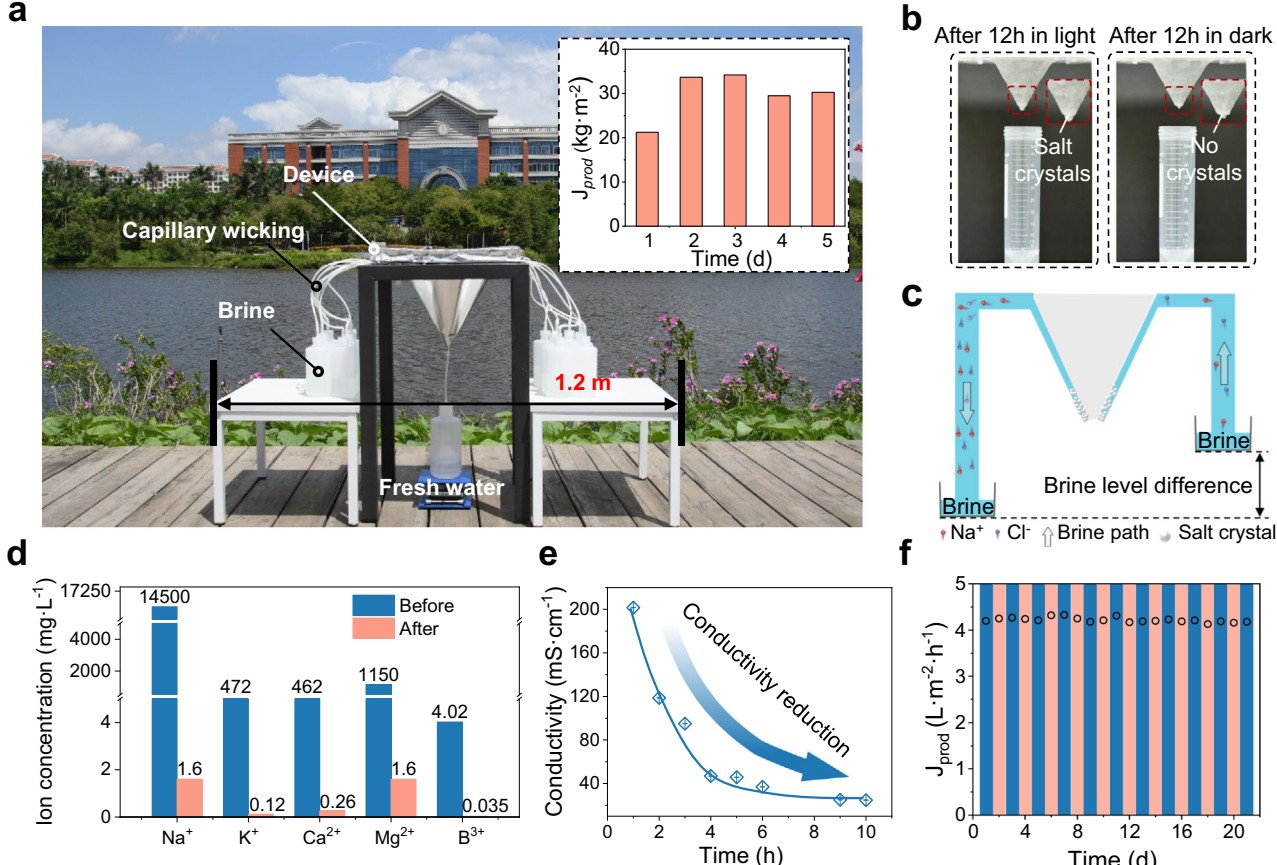

**Fig. 5 | The evaporation and salt resistance performance of asymmetric tapered eight-stage solar still. a** The photograph and water production of outdoor experiment of eight-stage device. **b** The salts dissolution process by setting water level difference. **c** The water and ion transport path of the device with water level difference. **d** The ion concentration before and after distillation. **e** The conductivity of the collected water at different time intervals. **f** The durability of the eight-stage device. A continuous 21-day (3-week) test under 1 kW·m⁻² solar intensity using 3.1 wt% natural seawater demonstrated stable performance, with the device consistently maintaining the water production of 4.13–4.33 L·m⁻²·h⁻¹. Error bars in **e** represent the s.d. ($n = 3$) and data are presented as mean values ± s.d.

cloudy day, and 5.9 L·m⁻²·d⁻¹ on an overcast day, underscoring its stability under varying weather conditions (Supplementary Figs. 17 and 18). To further validate the practical applicability of our design, we fabricated a scaled-up device with twice the original size. As shown in Supplementary Fig. 19, the scaled-up eight-stage device features a circular absorber with the opening diameter of 10 cm and an outermost layer with the diameter of 26 cm. We conducted outdoor tests on three sunny summer days, using natural sunlight during solar noon as the illumination source. The recorded light intensities are all above 0.75 kW·m⁻² (Supplementary Fig. 20a). As shown in Supplementary Fig. 20b, the water production ranged from 4.55 to 4.92 L·m⁻²·h⁻¹ across the 32 days, further demonstrating the real-world feasibility and robustness of the system.

Salt scaling poses a critical challenge for the long-term operation of the solar distillation system, especially in multistage device. Accumulated salt deposits on the evaporation layers are difficult to remove through conventional washing. Therefore, we introduced a water-level difference between the two supply channels, enabling efficient salt removal under dark conditions. As shown in Fig. 5b, salt crystals are observed on the evaporation layer at the tip of the cone after a full day outdoor operation. However, upon applying the water-level difference and leaving the device in darkness overnight, the salt crystals are significantly reduced or entirely eliminated. In addition, there are no contaminated water droplets observed in the collection tube, demonstrating the system's salt resistance performance (Fig. 5b). During salt removal process, water transports from the high-level side through the evaporation layer to the low-level side, dissolving salts

crystals along the way. Meanwhile, the dissolved salts are transported to the low-level side, where they are discharged (Fig. 5c). We set up a centrifuge tube on the low-level side for collecting the water discharged during the salt dissolution process. As water continuously flowed from the high-level side to the low-level side, the collected water volume increased over time (Supplementary Fig. 21). We then tested the conductivity of the collected salt water, as shown in Fig. 5e, where the conductivity gradually decreases, further verifying the effective dissolution and transport of salts path (Fig. 5e). This exceptional salt resistance enables stable device performance over 21-day (3-week) of operation, demonstrating the durability and practicality of the eight-stage system under real-world conditions (Fig. 5f).

## Discussion
In summary, we have proposed the ATMSMD system that simultaneously achieves ultrahigh vapor-to-water efficiency and water production through the systematic optimization of mass transfer equilibrium. Critical structural parameters, including the mass transfer gap and the ratio of condensation-to-evaporation areas, were fine-tuned to establish a comprehensive design principle for enhancing system performance. Extending this principle to a multistage configuration, we implemented the optimized eight-stage solar still with progressively decreasing mass transfer gap, achieving exceptional water production of 4.32 L·m⁻²·h⁻¹ and 34.2 L·m⁻²·d⁻¹ in outdoor conditions. This work, by opening up several highly promising fields, including the innovation of solar still structures and materials, the coupling with other water treatment technologies,

intelligent and automated control, and the development of distributed and water supply systems, will contribute to the sustainable utilization of global water resources. While in practical applications, the ATMSMD system still needs to address the issues such as increased fabrication and processing complexity associated with multilevel structures compared to single-stage configurations, as well as the necessity for more advanced design to facilitate efficient maintenance of such multistage configurations. The conceptual framework established in this work provides a guideline for the rational design of future passive multistage solar membrane distillation systems and demonstrates a performance advantage over previously reported systems.

## Methods

### Preparation of the device

The carbon nanotubes (CNTs) were synthesized by floating catalyst chemical vapor deposition (FCCVD). A quartz plate was used as the substrate. Xylene (Sinopharm Chemical Reagent Co., Ltd, ≥99.5%) was used as the carbon source and introduced at a rate of 6.5 mL·h⁻¹. Ferrocene (J&K, ≥99%) was dissolved in xylene (1 g per 30 ml) and acted as the catalyst. The carrier gas was argon (Linde, ≥99.99%) with flow rate of 250 cm³/min. The reaction was carried out over 3 h at 740 °C in a tube furnace (OTF-1200X, Hefei Ke Jing Materials Technology Co., Ltd) to obtain aligned MCNTs with a height of approximately 300 μm.

The absorption layer was prepared by mixing the Polydimethylsiloxane (PDMS, Sylgard 184, Dow Corning Corporation, ≥99.99%) and CNTs.

PDMS components A and B were uniformly mixed at a ratio of 10:1. The CNTs were then added to PDMS at a ratio of 0.1 g/8 g and stirred thoroughly. The mixture was placed in a vacuum drying oven to remove bubbles. The prepared mixture was evenly coated onto the surface of Al foil using a spin coating system (Spin-51, Shanghai Chemat Advanced Ceramics Technology Co., Ltd) and then dried in an oven at 70 °C for 2 h. The Al foil coated by CNTs/PDMS was then rolled into a funnel shape and secured with adhesive. A PDMS film was put onto the surface of the "funnel" to limit convection flow. A piece of fiber was cut into a semicircular shape and applied to the outside of the "funnel" as evaporation layer, with two attached supply channels.

Al foils (thickness: 0.05 mm) with a purity of 99.99% were thoroughly cleaned using an ultrasonic cleaning process with acetone (Sinopharm Chemical Reagent Co., Ltd, ≥99.5%) followed by deionized water. Following drying, the foils were immersed in a 1 mol·L⁻¹ aqueous solution of sodium hydroxide (Sinopharm Chemical Reagent Co., Ltd, ≥96%) for 30 to 60 seconds. Subsequently, they were sequentially rinsed with ethanol (Sinopharm Chemical Reagent Co., Ltd, ≥99.7%) and deionized water. Finally, the foils were blow-dried for subsequent use. A mixed aqueous solution comprising phosphoric acid (Shanghai Acmec Biochemical Technology Co., Ltd, ≥85%) and sodium fluoride was prepared and heated to a constant temperature of 70 °C in a water bath. Subsequently, the pretreated industrially aluminum foils were immersed in this solution for a duration of 5 min. Whereupon they were removed, sequentially cleaned with ethanol and deionized water, and then dried. Then, the aluminum foils treated with the mixed aqueous solution were immersed in liquid stearic acid (Shanghai Acmec Biochemical Technology Co., Ltd, ≥98%), which was maintained at 70 °C, for a period of 1 h. Following the soaking process, the foils were thoroughly rinsed in hot ethanol (70 °C) and subsequently cured in an oven at 80 °C for 30 min.

The modified Al foil was then rolled into a larger "funnel", with a layer of fiber applied to its exterior to serve as the evaporation layer for the next stage. Nine "funnels" with progressively increasing radius were fabricated, and then each of them was adhered to and fixed with PMMA plates. Finally, the nine "funnels" with PMMA plates were secured together using screws.

### Measurement and instruments

The simulated sunlight is provided by a solar simulator (China Education Au-light, CEL-HXF300). The optical power density was tested by a full spectrum intense optical power meter (CEL-NP2000). The temperature of the solar absorber was monitored using an infrared thermal imager (FLIR A325sc). The temperature of each evaporation layer was tested by a multichannel temperature collector (AnBai AT4708). The mass change of collected fresh water and vapor was tested by an electronic balance (Sartorius BSA224S). A spectrometer (UV-Visible/NIR Spectrophotometer UH5700, Hitachi) was used to measure the transmittance of the convection blocker and the absorbance of the light absorber. The photographs of device were taken by an optical camera (Nikon D850). The microscopic images were captured using a field emission scanning electron microscope (Zeiss, Gemini SEM 500). The ion concentration was tested by a conductively coupled plasma-optical emission spectrometry (ICP-OES, SPECTRO SPECTROBLUE FMX36). The contact angle was tested by a fully automatic contact angle and contour analysis instrument (OCA 100, Germany) and a liquid gating intelligent tester & analyzer (Gating Inspired Future Technology Co., Ltd, China, http://www.gift-xm.com/). The conductivity of the collected brine was tested by a conductivity meter (INESA, DZS-708L).

The complete vapor generation measurement procedure consists of four steps: (1) Pre-test preparation where we carefully seal both the brine tank and capillary wick with parafilm to prevent extraneous evaporation, while also sealing the device edges to minimize vapor leakage; (2) Controlled illumination phase where the fully prepared device undergoes 2 h illumination under solar simulator (1 kW·m⁻²); (3) Post-illumination measurements beginning with weighing the entire device (recorded as $m_0$), followed by disassembly to collect all condensate for separate weighing ($m_1$); and (4) Final verification where we thoroughly dry condensation surfaces before obtaining the final device mass ($m_2$). Through this rigorous protocol, we calculate the total vapor generation mass as $m_0 - m_2$ and the collected water production as $m_1$. The vapor production ($J$) and the water production ($J_p$) can be calculated by the followed equation:

$$J = \frac{m_0 - m_1}{t \times A} \tag{5}$$

$$J_p = \frac{m_2}{t \times A} \tag{6}$$

Where $A$ is the projected light area, and $t$ is the operation time. All indoor experiments were conducted at 25 °C and wind speed was excluded. Error bars represent the standard deviation of at least three independent measurements.

### COMSOL simulation

The vapor concentration field in the ATMSMD system are investigated by creating a transient model using the commercial software COMSOL Multiphysics® 6.0. The detailed process and parameters are provided in the supplementary information.

### Ray tracing simulation

The ray-tracing simulations were conducted using TracPro 7.0 software. Firstly, the corresponding physical model was built according to the light-collecting system used in the experiment. The optical simulation model of the is established by setting the solar incident model, the collector geometry parameters, optical parameters and the number of rays. Finally, the amplitude analysis of the simulation results is carried out to analyze the radial energy flux density distribution of the endothermic tube.

## Data availability
The data supporting the findings of this study are available within this article and its Supplementary Information. Source data are provided with this paper.

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

## Acknowledgements
This work was supported by the National Natural Science Foundation of China (Nos. 52025132, U24A20205, 21621091, 22021001, 22121001 awarded to X.H., and No. 52301087 awarded to B.Y.C.), the 111 Project (Nos. B17027 and B16029 awarded to X.H.), the National Science Foundation of Fujian Province of China (No. 2022J02059 award to X.H. and No. 2023J01783 award to B.Y.C.), and the New Cornerstone Science Foundation through the XPLORER PRIZE, awarded to X.H.

## Author contributions
X.H. conceived the idea. X.H. and W.H. designed the experiments. W.H. conducted the experiments. M.L. and J.C.W. designed the simulation. J.C.W. conducted a simulation. B.Y.C., Z.Y.F, and Y.Y.L. assisted in supplementing experimental data. X.Z. assisted in designing the structure of the device. L.Z. assisted in analyzing the data. W.H., J.C.W., Y.Y.L,. and X.H. write the original draft. X.H., W.H., and J.C.W. reviewed the draft.

## Competing interests

The authors declare no competing interests.
