## [Transparent Peer Review file · Nature Communications]

Asymmetric tapered multistage solar still with optimized mass transfer equilibrium for ultrahigh water production

Corresponding Author: Professor Xu Hou

Version 0:

Reviewer comments:

Reviewer #2

(Remarks to the Author)

In this study, the authors developed an optimized eight-stage passive solar distillation device, which achieved a total water yield of $4.32 \text{ L m}^{-2} \text{ h}^{-1}$ and an overall efficiency of 81% under 1 kW m^{-2} solar irradiation (using specific 3.1 wt% seawater).

The following are specific and detailed comments:

1. In the abstract, the authors state that a significant amount of vapor fails to condense because the evaporation and condensation areas are equal. This claim is incorrect. In currently reported evaporation-condensation systems, almost all condensation areas are larger than evaporation areas. In some cases, the condensation areas are even considerably larger than the evaporation areas, yet efficient condensation is still not realized, and the water production rate remains low.
2. The authors should conduct a comprehensive review of the existing literature on multistage solar distillation systems and integrate these advancements into the introduction, and make balanced comparison and discussion in the results part. Moreover, the limitations and advantages of the present study should be explicitly acknowledged.
3. From an application perspective, large-area evaporation membranes and large-area devices should be presented and included to demonstrate their practical value.

Version 1:

Reviewer comments:

Reviewer #2

(Remarks to the Author)

The authors claimed they fabricated a scaled-up eight-stage device. Unfortunately, I did not see the larger physical sample in the main text or the Supplementary Information.

Based on previous testing experience, small light receiving areas often yield calculated high evaporation rates; therefore, samples and devices with larger light receiving areas are more credible. From an application perspective, large-area devices should be presented and included to demonstrate their practical value.

The device shown in Figure 3 has a very small opening diameter of about 2.5 cm, which does not hold practical value for applications. It is suggested to present a device with a 10 cm opening diameter to validate the actual performance.

Version 2:

Reviewer comments:

Reviewer #2

(Remarks to the Author)

The revision is satisfactory for publication.

Responses to the reviewer's comments

For Reviewer 2:

Comment 1: *“In this study, the authors developed an optimized eight-stage passive solar distillation device, which achieved a total water yield of $4.32 \text{ L m}^{-2} \text{ h}^{-1}$ and an overall efficiency of 81% under 1 kW m^{-2} solar irradiation (using specific 3.1 wt% seawater). The following are specific and detailed comments: 1. In the abstract, the authors state that a significant amount of vapor fails to condense because the evaporation and condensation areas are equal. This claim is incorrect. In currently reported evaporation-condensation systems, almost all condensation areas are larger than evaporation areas. In some cases, the condensation areas are even considerably larger than the evaporation areas, yet efficient condensation is still not realized, and the water production rate remains low.”*

Reply 1: We thank the reviewer's comments on the accuracy of the abstract and appreciate the opportunity to improve our description. According to this comment, we revised the original phrasing as follows: *“...where vapor may not be condensed in time due to insufficient condensation capacity, or the available condensation capacity may be underutilized when evaporation is inadequate”*. As the reviewer pointed out, even in systems where the condensation area is significantly larger than the evaporation area, efficient condensation is often not achieved, and the water production remains low. However, in nearly all multistage configurations, the evaporation and condensation areas are typically designed to be equal, which also limits effective vapor condensation, especially in downstream stages with lower temperatures. These performance limitations are to a significant extent caused by the mismatch between evaporation and condensation capacities, which leads to incomplete vapor condensation or underutilized condensation potential. Therefore, the core of our study is to achieve optimal mass transfer equilibrium between evaporation and condensation processes to achieve maximum vapor and water production. The related enhancements are highlighted in yellow in the revised manuscript.

Comment 2: *“2. The authors should conduct a comprehensive review of the existing literature on multistage solar distillation systems and integrate these advancements into the introduction, and make balanced comparison and discussion in the results part. Moreover, the limitations and advantages of the present study should be explicitly acknowledged.”*

Reply 2: We thank the reviewer's suggestions on conducting comprehensive review of the existing literatures, strengthening comparative analyses, and in-depth discussion of the study's limitations and advantages. According to these suggestions, we conducted a comprehensive literature review on multistage solar distillation systems as follows: *“In recent years, multistage solar membrane distillation*

systems have attracted considerable attention due to their potential to recycle latent heat and enhance water production. Existing studies have improved system performance from multiple perspectives: (i) regulating heat transfer and implementing coupled heat–mass transfer models, enabling systematic control over heat transfer, vapor diffusion, and latent heat recovery within solar stills, thus breaking the efficiency limits of conventional passive systems; (ii) integrating multistage solar still with photovoltaic technologies to achieve water–electricity cogeneration and improve solar energy utilization; (iii) enhancing anti-scaling capabilities, enabling stable operation at salinities up to 20 wt% while maintaining high water production; and (iv) revealing how the optothermal characteristics of cover materials decisively govern heat–mass coupling in multistage solar stills, with a recent study proposing a material-thickness co-optimization paradigm that pushes efficiencies beyond 417% at low cost and high stability. Additionally, studies have proposed harnessing low-grade waste heat to drive evaporation in the final stages, further improving overall system efficiency through enhanced energy reutilization. These strategies have significantly enhanced the performance, stability, and integration potential of multistage solar membrane distillation systems. They have laid a solid foundation for understanding thermodynamic interactions of complex multistage configurations and provided valuable design insights for practical implementation. While these designs have greatly contributed to improving device performance, they primarily focus on optimizing either the evaporation or condensation process in isolation. To date, a systematic investigation of the mass transfer equilibrium between evaporation and condensation, which serves as the fundamental constraint governing vapor-to-water conversion, remains lacking. The mismatched evaporation and condensation capacity lead lots of vapor to fail to condense efficiently or the underutilization of condensation sites, constraining the further increase of water production and causing waste of solar energy”.

Through a more comprehensive literature analysis, we compared reported multistage solar membrane distillation systems with our optimized eight-stage device for treating seawater or 3.5 wt% NaCl solution. We plotted diagram of the water production for existing multistage devices and our device (Fig. R1, Fig. 3f), where our optimized eight-stage device exhibits the highest water production among the reported systems. The detailed parameters are listed in Table R1 (also Supplementary Table 3). We added a comprehensive comparison and discussion of the reported studies as follows: “To compare the optimized eight-stage device with existing multistage solar membrane stills, we plotted diagram of the water production for existing multistage devices and our optimized eight-stage device with treating natural seawater or 3.5 wt% NaCl solution (Fig. 3f). Detailed parameters from these studies, including light intensity, feedwater type, number of stages, and water production, have been compiled in Supplementary Table 3 for direct comparison. The results show that, for devices with different numbers of stages, our device consistently higher water production under comparable conditions. Notably, even when compared to eight- and ten-stage configurations, our optimized device also achieves the highest water production. We attribute this improvement not merely to structural adjustments but to a mechanism-driven design principle, which emphasizes achieving mass transfer equilibrium ($Q_{e,max} \approx Q_{c,max}$) between evaporation and condensation processes. This outcome supports the validity and generalizability of our proposed optimization strategy”.

Fig. R1 | Water production of optimized eight-stage device compared with previously reported multistage solar membrane distillation systems. Water production is benchmarked under the treatment of natural seawater or 3.5 wt% NaCl solution.

Table R1. The data of state-of-the-art multistage solar membrane distillation devices used for comparison.

Reference	Number of stages	Water production (L·m ⁻² ·h ⁻¹)	Feedwater type	Heat Flux (kW·m ⁻²)
Ref.20	3	2.23	Seawater	1
Ref.28	4	1.98	3.5 wt% NaCl	1
Ref.26	5	2.45	Seawater	1
Ref.38	6	2.2	Seawater	1
Ref.37	7	3.22	Seawater	1
Ref.39	8	2.25	3.5 wt% NaCl	1
Ref.25	8	2.63	Seawater	1
Ref.29	8	3.61	Seawater	1
This work	8	4.32	Seawater	1
Ref.27	10	3.82	3.5 wt% NaCl	1
Ref.25	10	3.17	Seawater	1
Ref.36	10	4.03	3.5 wt% NaCl	1

Furthermore, we added the explicit acknowledgment of limitations and advantages as follows “While in practical applications, the ATMSMD system still needs to address the issues such as increased fabrication and processing complexity associated with multilevel structures compared to single-stage configurations, as well as the necessity for more advanced design strategies to facilitate efficient maintenance of such multistage configurations. The conceptual framework established in this work provides a guideline for the rational design of future passive multistage solar membrane distillation systems and demonstrates a performance advantage over previously reported systems”.

The related enhancements are highlighted in yellow in the revised manuscript and supplementary information.

Comment 3: “3. From an application perspective, large-area evaporation membranes and large-area devices should be presented and included to demonstrate their practical value.”

Reply 3: We thank the reviewer’s suggestion on demonstrating the practical applicability of our device at larger scales. Based on this suggestion (present data based on a device with the diameter more than 10 cm), we fabricated a scaled-up eight-stage device with twice the original size, featuring a circular absorption area with a diameter of 10 cm. As the illumination spot of our solar simulator (China Education Au-light, CEL-HXF300) was not large enough to cover the enlarged light absorption area uniformly, we performed the tests outdoors under natural sunlight at noon on three representative sunny summer days. As shown in Fig. R2 (also Supplementary Fig. 19), the recorded light intensities on these days are all above 0.75 kW·m⁻². The corresponding water production, ranged from 4.55 to 4.92 L·m⁻²·h⁻¹, demonstrating that our system maintains high performance and stability even under realistic and larger-scale operational conditions. The related enhancements are highlighted in yellow in the revised manuscript and supplementary information.

Fig. R2 | The light intensity and water production at the noon of three sunny days. a, The light intensity of the three sunny days. **b,** The water production of the scaled-up device with twice the original size in the three days.

Responses to the reviewer's comments

For Reviewer 2:

Comment 1: “The authors claimed they fabricated a scaled-up eight-stage device. Unfortunately, I did not see the larger physical sample in the main text or the Supplementary Information. Based on previous testing experience, small light receiving areas often yield calculated high evaporation rates; therefore, samples and devices with larger light receiving areas are more credible. From an application perspective, large-area devices should be presented and included to demonstrate their practical value. The device shown in Figure 3 has a very small opening diameter of about 2.5 cm, which does not hold practical value for applications. It is suggested to present a device with a 10 cm opening diameter to validate the actual performance.”

Reply 1: We thank the reviewer for the suggestion to present a larger physical sample. According to this suggestion, we added photographs of the scaled-up eight-stage device in outdoor test, as well as front and side views of the disassembled structure. As shown in Fig. R1 (also Supplementary Fig. 19), the device features an opening diameter of 10 cm and an outermost layer with the diameter of 26 cm. The outdoor tests conducted at noon on sunny summer days with the light intensity above 0.75 kW/m^2 . In addition, we would like to clarify a possible misunderstanding regarding the device size shown in Fig. 3a. The actual opening diameter of the device is 5 cm, not 2.5 cm. This misunderstanding may have resulted from the scale bar in the earlier version not being sufficiently accurate. In the revised manuscript, we have improved the accuracy of the scale bar and added an illustration to explicitly label the actual device size (Fig. R2, also Fig. 3a). The related enhancements are highlighted in yellow in the revised manuscript and supplementary information.

Fig. R1 | The photographs of scaled-up eight-stage device. a, The photograph of outdoor test at the noon of sunny summer day. The illustration shows the light intensity. **b,** The front view of the

disassembled structure of the scaled-up eight-stage device. **c**, Side view of the disassembled structure of the scaled-up eight-stage device.

Fig. R2 | The photograph of the device with a 5 cm opening diameter. This device was used in laboratory experiments to optimize A_c/A_e for each stage. The design of the larger-scale device (10 cm diameter) was based on these optimized parameters to ensure scalability and practical applicability. The scale bar of the enlarged image is 1 cm.

We thank the reviewer's suggestion, which is valuable in improving the quality of the work.